# APPLICATION OF METRIC TRANSFORMATION IN ONE-STEP RETROSYNTHESIS

## Abstract

In this article, we investigate the impact of Deep Metric Learning and Transformer architecture on predicting the retrosynthesis of Simplified Molecular Input Line Entry System (SMILES) chemical compounds.

We demonstrate that combining the Attention mechanism with Proxy Anchor Loss is effective for classification tasks due to its strengths in capturing both local and global contexts and differentiating between various classes.

Our approach, which requires no prior chemical knowledge, achieves promising results on the USPTO-FULL dataset, with accuracies of 53.4%, 83.8%, 90.6%, and 97.5% for top-1, top-5, top-10, and top-50 predictions, respectively.

We further validate the practical application of our approach by correctly predicting the retrosynthesis pathways for 63 out of 100 randomly selected compounds from the ChEMBL database and for 39 out of 60 compounds selected by Bayer's chemists and from PubChem.

## 1 INTRODUCTION

Designing molecules and materials with desired properties is a practical goal of chemistry and materials science. Chemoinformatics involves the use of data and computational methods to investigate and comprehend the connections between molecular structures and their properties, paving the way for the discovery of novel functional molecules.

Chemists prioritize designing synthesis pathways that generate target molecules through a sequence of chemical reactions. A common approach, known as retrosynthetic analysis, involves deconstructing target molecules into precursor molecules and then further deconstructing these precursors until all are available in the stock database for synthesis.

Retrosynthetic planning involves two primary tasks: single-step retrosynthesis prediction and multi-step retrosynthetic planning.

Single-step retrosynthesis is treated as a prediction problem, where the input is a given product molecule and the output is the predicted set of reactant molecules.

Multi-step retrosynthesis planning aims at producing a relizable synthetic route for a specific compound by breaking it down into simpler intermediates or precursors. This process is carried out through retrosynthetic analysis, where the desired compound is iteratively deconstructed until one of the stopping criteria is met: identifying known or purchasable building blocks, reaching a time limit, reaching a maximum tree depth, or fulfilling other predefined criteria (Samuel et al. 2020). Multi-step retrosynthesis planning is approached as a search problem, guided by solutions derived from single-step retrosynthesis.

There are two main approaches to the one-step retrosynthesis problem: template-based methods and template-free methods.

Template-based methods mimic chemists' reasoning by using a pool of reaction templates to identify potential reaction centers. The goal is to find the appropriate template that allows the deconstruction of product molecules into their reactant molecules. Template-based methods are a classical approach, where the one-step retrosynthesis problem is considered a classification problem. This approach is reliable since it uses experts' knowledge about chemical reactions and mimics the way chemists work, which makes it easy to use for experts (Meng et al. 2023). This is also the approach for our experiments in this paper.

Template-free methods, on the other hand, are considered sequence-to-sequence learning processes, where the input consists of string-like representations of chemical products (SMILES), and the output corresponds to the SMILES strings of reactants (Meng et al. 2023). The most common models used in this approach are Transformer-based (Vaswani et al. 2017).

# 2 METHODS AND DATASETS

## 2.1 DATASET

### 2.1.1 USPTO, AIZYNTHFINDER, AIZYNTHTRAIN, REACTION UTILS AND RXNMAPPER

USPTO-FULL is a dataset extracted from chemical reactions in US patents (1976-Sep 2016) (Lowe 2017). It consists of approximately 3.5 million reactions with almost 1.2 million reaction templates.

Several free and open-source software tools are available for downloading and preprocessing the USPTO-FULL reaction dataset. We utilize AiZynthTrain (AZT) (Genheden, Norrby, and Engkvist 2022), Reaction Utils (Kannas et al. 2022), and RXNMapper (Schwaller et al. 2021) to prepare and train our one-step retrosynthesis model due to their robustness and comprehensive end-to-end pipelines.

RXNMapper and Reaction Utils are used for downloading, preparing, and performing atom mapping on the USPTO dataset. This is followed by AiZynthTrain, which provides a collection of routines, configurations, and pipelines for transforming the atom-mapped dataset into molecular fingerprints[1], splitting the data, and training one-step retrosynthesis prediction models. These pipelines also generate human-readable reports, covering all stages from data preprocessing to the training process.

On the other hand, AiZynthFinder (AZF) (Samuel et al. 2020) is a free and open-source software for multi-step retrosynthetic planning. Its primary search algorithm is based on Monte Carlo Tree Search (MCTS) (Browne et al. 2012), which recursively breaks down a target molecule into purchasable precursors. We use AiZynthFinder and its MCTS to conduct our experiments on multi-step retrosynthesis tasks. Additionally, we incorporate Nested Monte Carlo Search (NMCS) (Tristan Cazenave 2009), which has already demonstrated promising results in multi-step retrosynthesis (Roucairol and T. Cazenave 2024), in order to gain a deeper understanding of the interaction between one-step retrosynthesis models and search algorithms in multi-step retrosynthesis problems.

### 2.1.2 DATASET

Our primary objective is to enhance the performance of our one-step retrosynthesis model to subsequently improve multi-step retrosynthesis planning with AiZynthFinder (Samuel et al. 2020). This objective builds on the work of comparing search algorithms in multi-step retrosynthesis (Roucairol and T. Cazenave 2024), including Nested Monte Carlo Search (Tristan Cazenave 2009) and Greedy Best First Search (Doran and Michie 1966). To achieve this, we aim to design a robust one-step retrosynthesis architecture that can deliver satisfactory performance within defined time constraints for inference, in order to efficiently support multi-step retrosynthesis planning.

Therefore, we use the USPTO-FULL dataset (as opposed to the USPTO-50k dataset (Schneider, Stiefl, and Landrum 2016) used in other studies). This dataset was downloaded and prepared using RXNUtils and RXNMapper, and subsequently filtered with AiZynthTrain. To ensure a fair comparison with the AiZynthTrain base model and its associated experiments, we applied all default filters provided by AiZynthTrain (Genheden, Norrby, and Engkvist 2022), including the criterion to retain only templates with more than $N = 3$ example reactions in the USPTO-FULL dataset.

After filtering, the dataset is divided into two subsets: 812,948 samples associated with 42,134 unique reaction templates for the one-step retrosynthesis model, and a "ring-breaker" dataset containing 57,227 ring reactions with 5,225 reaction templates, which is used to train the ring-breaker model (Thakkar et al. 2020). The one-step retrosynthesis dataset is split into 658,907/97,776/56,265

---

[1]A numerical representation of the molecule, mapping it to a sparse discrete representation space. Molecular fingerprints provide a computationally efficient and consistent way to represent the chemical properties (structural, physicochemical, etc.) of large-scale chemical datasets (Yang et al. 2022).

for training/validation/testing. The ring-breaker dataset is split as 48,800/4,044/4,383 for training/validation/testing.

AiZynthTrain also transformed each molecule into its molecular fingerprint, which serves as the input for the models. The associated labels are the reaction templates that need to be applied to deconstruct the molecules into smaller reactants.

A key feature of the USPTO-FULL dataset is the large number of reaction templates (or classes) and the significant imbalance between them, with sample counts ranging from 1 for the smallest templates to nearly 8,000 for the largest. This imbalance makes training more difficult, especially for underrepresented classes.

For a clearer assessment with smaller datasets, we also performed experiments on the USPTO-50k dataset (Schneider, Stiefl, and Landrum 2016). After AiZynthTrain filtering, we obtained 27,704/4,580/2,376 samples for training/validation/testing, along with a negligible ring-breaker dataset. Like USPTO-FULL, the filtered USPTO-50k is heavily imbalanced, with nearly half the templates having just 1-3 training samples, making it prone to overfitting.

## 2.2 METHODS

### 2.2.1 BACKBONE MODEL

**Deep Metric Learning and Proxy Anchor Loss**

Despite their extensive use in supervised deep-learning applications, including one-step retrosynthesis problems, cross-entropy-based loss functions are often less effective when there is significant intra-class variance and minimal inter-class variance within the input data distribution.

Deep Metric Learning aims to measure the similarity between data samples by learning a representation function that maps these samples into an embedding space, where samples from the same class are closely grouped together.

The quality of an input's representation largely depends on the loss functions used to train the networks. These loss functions are typically categorized into two classes: pair-based and proxy-based.

Pair-based losses are derived from pairwise distances between data points in the embedding space. A well-known example is Contrastive Loss (Bromley et al. 1993), which aims to minimize the distance between pairs of data points with identical class labels and maximize the distance between those with different labels. These losses provide rich data-to-data information by directly comparing data points. However, this approach results in extremely high training complexity and requires a specific arrangement of data to generate both negative and positive pairs.

Proxy-based losses address this issue by introducing proxies as representatives, which are learned as part of the network parameters. Each data point is associated with proxies, typically one per class, enabling the model to leverage data-to-proxy relationships. In essence, a proxy can be thought of as a learnable centroid for each class. The model's task is to adjust the representations of both samples and proxies so that each sample is close to its corresponding proxy, each proxy is close to its respective samples, and optionally, the distance between samples of different classes is maximized. Examples of such losses include Proxy NCA (Movshovitz-Attias et al. 2017) and Proxy Anchor Loss (Kim et al. 2020).

We use Proxy Anchor Loss to train our backbone model because it captures both data-to-proxy and data-to-data relationships by taking each proxy as an anchor and associating it with the entire data in a batch. The loss is given by (Kim et al. 2020)

$$
\mathcal{L}(X) = \frac{1}{|P^+|} \sum_{p \in P^+} \log \left( 1 + \sum_{x \in X_p^+} e^{-\alpha(s(x,p)-\delta)} \right)
$$
$$
+ \frac{1}{|P|} \sum_{p \in P} \log \left( 1 + \sum_{x \in X_p^-} e^{\alpha(s(x,p)+\delta)} \right)
\tag{1}
$$

In equation 1, $\delta > 0$ is a margin, $\alpha > 0$ is a scaling factor, $P$ indicates the set of all proxies, and $P^+$ denotes the set of positive proxies of data in the batch. Also, for each proxy $p$, a batch of embedding vectors $X$ is divided into two sets: $X^+$, the set of positive embedding vectors of $p$, and $X^- = X - X^+$ (Kim et al. 2020).

One issue we observed with Proxy Anchor Loss during training is the potential for gradient instability due to the large number of unique reaction templates. Therefore, careful selection of hyperparameters, particularly $\alpha$, is crucial. We experimented with $\alpha$ values ranging from 1 to 128. With small $\alpha$ values such as 1, 2, and 4, the interaction between samples and proxies was too weak, while larger values such as 64 and 96 led to overly aggressive interactions, causing gradient instability. We found that $\alpha = 32$ yielded optimal results, avoiding gradient issues, consistent with the findings in the Proxy Anchor Loss paper (Kim et al. 2020).

**Attention and Positional Embedding**

Transformers (Vaswani et al. 2017) represent the State-of-the-Art in Natural Language Processing due to their ability to capture both local and global context. The Transformer architecture has also achieved notable success in Computer Vision, as demonstrated by the Vision Transformer (ViT) (Dosovitskiy et al. 2021).

Drawing inspiration from both the original Transformer and the Vision Transformer, we conceptualize a chemical compound as a sentence composed of words. Consequently, we transform the input into vectors of consistent size to apply multi-head self-attention, the mechanism used in Transformer model

We treat a molecule as a sentence, where the position of each 'word' is crucial for the model's effective interpretation. Thus, we incorporate Positional Embedding before applying the attention mechanism.

**Metric Transform**

We refer to our backbone model, which incorporates Positional Embedding, the Attention mechanism of the Transformer, and Proxy Anchor loss, as the Metric Transform (see Fig 2). The output of the final Attention layer is passed through a Global Max-Pooling 1D layer to capture the most significant information about the input molecule. The Max-Pooling operation reduces the spatial dimensions of the feature maps by selecting the maximum value within a specified spatial window, thereby helping to mitigate overfitting. Additionally, one or more fully connected layers can be added after Max-Pooling to improve the model's ability to distinguish between molecules and facilitate fine-tuning.

This model is then trained using Proxy Anchor Loss to enhance its ability to distinguish between samples from the same class and samples from different classes.

Typically, proxies are initialized and trained with a high learning rate on a pre-trained model, such as ResNet (He et al. 2016) or ViT. This approach allows the proxies to quickly adapt and subsequently refine the model parameters to enhance performance. In contrast, our approach involves training the backbone model from scratch. Consequently, we initialize and train the proxies using the same learning rate as the backbone model to encourage the model's parameters to be optimized at a similar pace to the proxies, instead of letting the proxies take the lead as in the original implementations. We also experimented with using a higher learning rate for the proxies and, conversely, a lower learning rate for the proxies. In both cases, the results were less attractive[2] compared to using the same learning rate for both the proxies and the model's parameters.

### 2.2.2 SUBCLASS MAPPING AND FINE-TUNING PROCESS

Following a traditional fine-tuning approach, we enhanced our backbone model by adding one or more fully connected layers. This resulted in a 3-4% improvement in performance over the AiZynthTrain base model, demonstrating satisfactory progress.

Unlike moderate-sized classes, which tend to form well-defined clusters during the representation learning process, samples from the major classes exhibit only localized clustering. While our ap-

---

[2]By "attractive," we refer to the capability of distinguishing samples from minor classes, as the major classes can be handled by a solution we propose below.

proach improves the model's predictive performance compared to the base model, the inherent bias of deep neural networks toward the major classes restricts the model's ability to generalize effectively to rare templates.

We address this problem by performing "subclass mapping" (see Fig 1) for all major classes using k-means clustering (Macqueen 1967). Instead of treating a given major class as a single entity, we divide it into smaller subclasses, such that samples within each subclass are closer to one another. This approach has two main benefits:

1. Reduces the imbalance between major and minor classes.

2. Makes it easier for the model to distinguish similarities and differences by using subclass labels instead of whole-class labels.

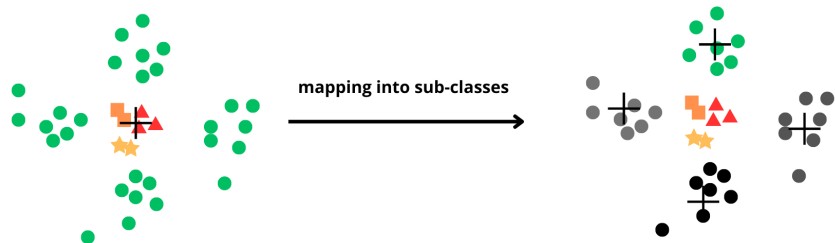

Figure 1: Schematic diagram of the subclass mapping process. Although the green points are locally clustered, their centroid is close to samples from minor classes, which makes distinguishing between classes less effective. We use the subclass mapping to address this issue.

After this step, the number of classes will increase compared to the original dataset. Despite the greater number of classes, this approach helps the model fine-tune more effectively because samples within each subclass are now closer to each other through their local cluster than they were within the original major class. Additionally, the class imbalance will be less severe.

We also define a custom layer that remaps subclasses back to their original classes, enabling reaction template predictions for a given molecule. In the case of a reaction template (class) being split into two or more subclasses, the remapping layer will use predefined criteria to determine the probability of predicting the original class.

Next, we fine-tune our backbone model by adding a fully connected layer to perform the classification task using traditional cross-entropy loss with subclasses and incorporate the remapping layer to remap the subclasses back to their original classes (see Fig 3).

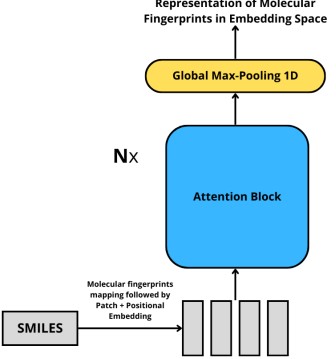

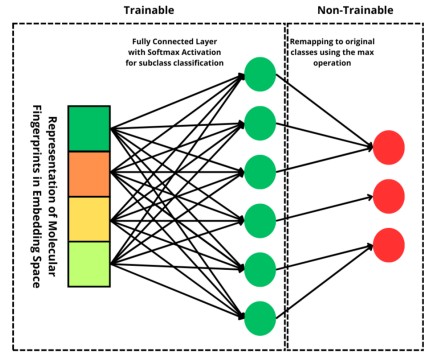

Figure 2: Diagram of the Metric Transform.          Figure 3: Diagram of the fine-tuning process.

# 3 RESULTS AND DISCUSSION

## 3.1 ONE-STEP RETRO-SYNTHESIS

As described above, our primary objective is to develop a robust model that can be fine-tuned for downstream tasks and applied to other template-based architectures to enhance their performance. Therefore, we trained both our one-step and ring-breaker models using datasets derived from the USPTO-FULL dataset. Each model was trained solely on its training set, with its validation set used for model selection and early stopping criteria..

The entire training process, which included training the backbone model, subclass mapping, and fine-tuning on USPTO-FULL, required only a few hours in total on a 3.30 GHz AMD Ryzen 5 6600H Linux machine with 16 GB of RAM and a NVIDIA GeForce RTX 3050 4 GB GPU.

**Comparison between AiZynthTrain base model and Metric Transform**

We first compare our architecture with the AiZynthTrain base model for one-step retrosynthesis and the ring-breaker model. It is important to note that the ring-breaker model is trained similarly to the one-step model but on the ring-breaker dataset, a derived dataset that contains only ring reactions.

| Model | Top-1 Accuracy | Top-5 Accuracy | Top-10 Accuracy | Top-50 Accuracy |
|---|---|---|---|---|
| AZT | 51.1/52.4 | 77.3/81.8 | 83.7/87.9 | 92/95.1 |
| MetricTransform (Ours)[a] | 54.2/51.9 | 81.4/83.2 | 87.3/90.4 | 95/97.3 |
| MetricTransform (Ours) | 55/53.4 | 81.9/83.8 | 88.1/90.6 | 95.1/97.5 |
| AZT ring-breaker | 70/70.8 | 90.4/91 | 93.6/94 | 97.7/98.2 |
| MetricTransform ring-breaker (Ours)[a] | 74.3/74.5 | 94.9/95.2 | 97.5/97.9 | 99.4/99.5 |
| MetricTransform ring-breaker (Ours) | 74.3/74.4 | 95/95.4 | 97.5/97.7 | 99.5/99.5 |

[a] Results without the subclass mapping process.

Table 1: Top-k accuracy of the AiZynthTrain base model and Metric Transform on the USPTO-FULL dataset filtered by AiZynthTrain. Accuracies are reported as the left side of the slash for the validation set, and the right side for the test set.

**Comparison of Models Based on Predictive Capacity and Inference Speed**

State-of-the-Art models, such as GLN (Dai, Li, C. Coley, et al. 2019) and the Augmented Transformer (Tetko et al. 2020), utilize the curated USPTO-FULL dataset (Dai, Li, C. Coley, et al. 2019). This curated version of the original USPTO-FULL dataset (Lowe 2017) was created by removing duplicate reactions and those with incorrect atom mappings, resulting in training/validation/testing sets with 800k/100k/100k samples, respectively.

On the other hand, using the default filtering and splitting of AiZynthTrain, we have dataset of 707,707/101,820/60,648 samples for training/validation/testing. These datasets are split for one-step modeling and ring-breaker modeling. To enable a fair comparison with other models on the USPTO-FULL benchmark, we rescaled our model's performance by applying the following rules:

- Because our models are trained solely on the training set, we rescale accuracies as the average of weighted accuracies from both the one-step model and the ring-breaker model, on both the validation and testing sets.

- Since the ratio between the training set and the validation/testing set is 4.36 (707,707:162,468) instead of the typical 4.0 (696,140:174,035), as used in other papers, this discrepancy implies that some samples should belong to the validation/testing set rather than the training set. Therefore, we consider these 11,567 samples as failed predictions, even though this may not necessarily reflect the actual training and inference process.

- Additionally, we have 25,965 fewer samples[3] compared to the validation/testing sets used in the curated USPTO-FULL dataset (Dai, Li, C. Coley, et al. 2019). This shortfall includes samples that belong to rare reaction templates, which only appear once or twice in the entire

---

[3]This shortfall is calculated based on the assumption that the total number of samples in the validation/testing sets is 174,035, with 11,567 samples immediately considered as failed predictions because they are actually part of the training set.

dataset and have been filtered out by AiZynthTrain. Consequently, we assume failure to predict these samples, similar to the approach used in the Augmented Transformer (Tetko et al. 2020).

Furthermore, in the retrosynthesis problem, the primary function of a single-step model is to serve as an environment during the multi-step search process. Therefore, inference time is a crucial metric for evaluating these models, in addition to predictive performance. Faster single-step models can enable more extensive searches within limited time and computational resources (Maziarz et al. 2023). Despite its importance, inference speed is often overlooked in existing studies. Our work addresses this gap by evaluating models based on their inference time and comparing them with other available models, offering a comprehensive view of inference speed in one-step retrosynthesis.

We present the following comparison of the performances in both predictive capacity and inference speed of models on the USPTO-FULL dataset in Table 2 and 3.

| Model | Top-1 Accuracy | Top-10 Accuracy |
|---|---|---|
| Retrosim[a] | 32.8 | 56.1 |
| Neuralsym[a] | 35.8 | 60.8 |
| GLN[a] | 39.3 | 63.7 |
| Augmented Transformer[b] | 44.4 | 70.4 |
| AZT[c] | 42.7 | 69.6 |
| MetricTransform (Ours)[c] | 45.1 | 72.7 |

[a] Results for Retrosim (C. W. Coley, Rogers, et al. 2017), Neuralsym (Segler and Waller 2017), and GLN as reported by Dai, Li, C. W. Coley, et al. 2020.
[b] Results reported by Tetko et al. 2020 by assuming that Augmented Transformer failed for all 4% of excluded reactions from the curated USPTO-FULL dataset.
[c] Recalculated results on the filtered USPTO-FULL dataset by AiZynth-Train

Table 2: Rescaled Top-k Accuracy of Various Models on the USPTO-FULL Dataset

| Model | Inference Speed (sec/sample) |
|---|---|
| GLN[a] | $10^{-1}$ |
| Transformer[a] | $10^{-1}$ |
| MHNreact[a] | $10^{-3}$ |
| NeuralSym[a] | $10^{-3}$ |
| AZT[b] | $10^{-4}$ |
| MetricTransform (Ours)[b] | $10^{-4}$ |

[a] Results for GLN, Transformer, MHNreact, and NeuralSym as reported by Seidl et al. 2022 using Nvidia GPUs (Titan V 12GB, P40 24 GB, V100 16GB, A100 20GB MIG).
[b] Average inference time obtained by evaluating on the whole testing set consisting of 56,265 samples with batch size = 1, using an NVIDIA GeForce RTX 3050 4 GB GPU.

Table 3: Inference speeds of various models, expressed in terms of base-10 exponents

## Coverage

USPTO datasets have a key characteristic: some reaction rules are absent from the training set. Therefore, the percentage of coverage is often considered the theoretical upper limit for model performance. To better assess predictive capacity, we re-evaluate model performance based on the total coverage of reaction templates, providing a clearer view of each model's learning capabilities.

| Model | Coverage | Top-1 Accuracy | Top-5 Accuracy | Top-10 Accuracy | Top-50 Accuracy |
|---|---|---|---|---|---|
| GLN[a] | 93.3 | 56.3 | 81.0 | 89.7 | 99.0 |
| GLN[b] | 93.3 | 68.8 | 91.3 | 96.5 | 99.9 |
| LocalRetro[a] | 98.1 | 54.4 | 87.6 | 94.2 | 99.6 |
| LocalRetro[b] | 98.1 | 65.1 | 94.2 | 98.2 | 99.8 |
| LocalRetro[c] | 97.0 | 55.8 | 81.8 | 87.0 | 93.2 |
| AZT[d] | 100 | 52.6 | 79.6 | 85.7 | 93.4 |
| MetricTransform (Ours)[d] | 100 | 55.5 | 83.2 | 89.5 | 96.1 |

[a] Results on USPTO-50k; reaction class unknown.
[b] Results on USPTO-50k; reaction class known.
[c] Results on USPTO-MIT (Jin et al. 2017).
[d] Results on USPTO-FULL.

Table 4: Rescaled top-k Accuracy of Various Models based on template coverage

## USPTO-50k

We conducted quick experiments with the filtered USPTO-50k dataset and compared the AiZynth-Train base model with our approach. Although the dataset is less imbalanced than USPTO-FULL, many templates with only 1-3 samples might complicate the training process and increase the risk of overfitting. Initially, we used Attention layers but found that they caused severe overfitting. Therefore, we replaced them with a fully connected layer in the Metric Transform model and then followed the same process as for USPTO-FULL. Due to the lesser imbalance, models with and without sub-

class mapping showed no significant performance difference. The results were obtained without knowledge of reaction type.

| Model | Top-1 Accuracy | Top-5 Accuracy | Top-10 Accuracy | Top-50 Accuracy |
|---|---|---|---|---|
| O-GNN | 54.1 | 86.0 | 92.5 | 98.3 |
| LocalRetro | 53.4 | 85.9 | 92.4 | 97.7 |
| GLN | 52.5 | 75.6 | 83.7 | 92.4 |
| AZT[a] | 43.5/44.3[b] | 70.2/73.9 | 77.1/81.5 | 88.7/90.9 |
| MetricTransform (Ours)[a] | 48/46.4 | 75.8/77 | 82.6/84.9 | 92.4/94 |

[a] Results obtained with filtered USPTO-50k
[b] Results on validation/testing set.

Table 5: Top-k accuracy of various models on the USPTO-50k dataset (reaction class unknown)

**Discussion of One-Step Retrosynthesis Models**

Throughout our experiments, we demonstrate that a combination of the attention mechanism, deep metric learning, and subclass mapping can achieve satisfying predictive performance comparable to state-of-the-art methods, such as GLN and Augmented Transformer (see Table 2), on the USPTO-FULL dataset while maintaining significantly faster inference times (see Table 3).

Although we achieved good results compared to the AiZynthTrain base model on the filtered USPTO-50k dataset, we observed limitations in our approach with smaller datasets compared to other models such as GLN, LocalRetro, and $\mathcal{O}$-GNN (Zhu et al. 2023) (see Table 5).

On the other hand, the experiments (see Table 2 and 4) demonstrate that with larger datasets, state-of-the-art models like GLN and LocalRetro tend to exhibit slightly reduced performance. In contrast, our approaches show improved performance on larger datasets.

We also further investigate subclass mapping and its potential in Section A.1.

### 3.2 MULTI-STEP RETRO-SYNTHESIS

#### 3.2.1 MULTI-STEP RETROSYNTHESIS

We evaluated our model on multi-step retrosynthesis on a 3.30GHz AMD Ryzen 5 6600H, Linux machine with 16 Gb of RAM, NVIDIA GeForce RTX 3050 4GB, using 2 subsets:

- 100 compounds randomly selected from the ChEMBL database (Samuel et al. 2020).
- 60 molecules[4], consisting of 20 molecules selected by Bayer's chemists (molecules ID starting with A) and 40 molecules randomly selected from PubChem (Ertl and Schuffen-hauer 2009), chosen to cover a range of molecular sizes from small to large (molecules ID starting with C).

The first subset was designed to facilitate a comparison between AiZynthFinder and ASKCOS, and since both tools use Monte Carlo Tree Search, our tests on this subset will exclusively use this search algorithm.

The second subset, initially intended to test the predicted difficulty of retrosynthesizing these molecules, features some of the hardest compounds to synthesize, according to Bayer's chemists and Ertl's team. Therefore, we utilize both Monte Carlo Tree Search and Nested Monte Carlo Search to examine the retrosynthetic capacity of the combination of our model with different search algorithms.

We then compare our multi-step retrosynthesis performance with ASKCOS (C. W. Coley, Barzilay, et al. 2017) and Depth-First Proof-Number Search (DFPN) (Franz et al. 2022). We note the significant differences among them:

- AiZynthFinder and Metric Transform use the USPTO-FULL dataset with the ZINC database (Sterling and Irwin 2015) as stock.

---

[4]Dataset available at `https://doi.org/10.5281/zenodo.6511731`

- ASKCOS uses the Reaxys dataset (*Reaxys* 2019) with Sigma Aldrich and eMolecules (C. W. Coley, Thomas, et al. 2019) as stock .

- DFPN uses several public and non-public data sources, consisting of 8,616,239 molecules and 270,605 reaction templates obtained after extensive data cleaning and preprocessing, with chemicals labeled as buyable according to the suppliers for Bayer Research as stock (Franz et al. 2022).

We observe that among these datasets, the USPTO-FULL and ZINC are considerably smaller compared to Reaxys, Sigma Aldrich, and eMolecules. DFPN utilizes a significantly larger dataset, which is ten times the size of USPTO-FULL, along with an internal stock database. However, because DFPN relies on non-public datasets, we are unable to explore their resources further.

**Comparison on 100 molecules between ASKCOS, AiZynthFinder and Metric Transform**

To ensure a fair comparison, we maintain the original AiZynthFinder setup (Samuel et al. 2020). In addition to the number of solved molecules, which serves as the primary criterion in multi-step retrosynthesis, we also present additional statistics to provide a deeper understanding of how the one-step retrosynthesis model impacts solving capacity

| Model | Ring Breaker | Iter = 100 | Iter = 10,000 | Solved |
|---|---|---|---|---|
| ASKCOS | | | | 62 |
| AZF (Original Result) | X | X | | 55 |
| AZF (Our Test) | | X | | 50 |
| AZF (Our Test) | | | X | 59 |
| AZF (Our Test) | X | X | | 52 |
| AZF (Our Test) | X | | X | 60 |
| MetricTransform (Ours) | | X | | 57 |
| MetricTransform (Ours) | | | X | 63 |
| MetricTransform (Ours) | X | X | | 55 |
| MetricTransform (Ours) | X | | X | 63 |

Table 6: Comparison of different models and their performance with various configurations and parameters. The table presents results for models with and without the ring-breaker feature, and across different iteration limits, within a time limit of 120 seconds per molecule.

| Model | Average Solution Time | Average Number of Steps | Average Number of Precursors | Solved |
|---|---|---|---|---|
| ASKCOS | 14.3 | 3.3 | 3.2 | 62 |
| AZF (Original Result) | 7.1 | 2.4 | 2.7 | 55 |
| AZF (Our Test) | 7.7 | 3.0 | 3.5 | 60 |
| MetricTransform (Ours) | 6.0 | 2.8 | 3.4 | 63 |

Table 7: Comparison of retrosynthesis models based on key performance statistics. Results are shown for models with a ring-breaking feature, limited to 120 seconds and 10,000 iterations.

**Comparison on 60 molecules between DFPN, AiZynthFinder and Metric Transform**

For the AiZynthFinder MCTS experiment, we first set the cutoff number (the maximum number of possible moves returned for a molecule) to 5 with time limits of 300 and 600 seconds, then increased the cutoff to 50 with limits of 900 and 1200 seconds.

In the NMCS experiment, we also utilize `nmcs1top5` (level 1 NMCS with only the 5 best moves from each state), followed by `nmcs1top50` and `nmcs2top50` (Roucairol and T. Cazenave 2024) for more expensive and exhaustive search.

For both experiments, we set the max-transforms (the maximum depth of the search tree) to 7, consistent with the setting used for DFPN.

| Model | MCTS[a] | NMCS | MCTS[b] | DFPN |
|---|---|---|---|---|
| AZF model[c] | 31 | 36 | | |
| DFPN model[d] | | | 38 | 41 |
| Metric Transform (Ours) | 33 | 39 | | |

[a] AiZynthFinder Monte Carlo Tree Search.
[b] Depth-First Proof-Number Search Monte Carlo Tree Search.
[c] Reported results by Roucairol and T. Cazenave 2024.
[d] Reported results by Franz et al. 2022.

Table 8: Comparison of retrosynthesis models using different search algorithms

A comprehensive analysis of the time taken to solve each molecule can be found in Section A.3

**Discussion on Multi-Step Retro-Synthesis**

We demonstrate the predictive performance of our approach by using different search algorithms on various subsets. Our models yield results comparable to ASKCOS and slightly below DFPN (see Table 6 and 8), despite utilizing a considerably smaller dataset, a reduced stock database, and limited computational resources. Additionally, we reaffirm the potential of NMCS in multi-step retrosynthesis planning, as demonstrated in previous works (Roucairol and T. Cazenave 2024).

It is important to note that both the dataset for one-step retrosynthesis modeling and the stock database are crucial in multi-step retrosynthesis planning. A larger dataset can encompass a broader range of reaction templates, while an extensive stock database enhances the flexibility of retrosynthesis planning, resulting in significantly improved outcomes (Samuel et al. 2020).

The ring-breaker model yields mixed results, which are discussed in Section A.2.

## 4 CONCLUSION

The combination of deep metric learning, attention mechanisms, and subclass mapping resulted in a robust architecture that addresses the inherent imbalance in chemical reaction datasets. This approach demonstrates strong performance across both common and rare reaction types, delivering satisfactory results in both one-step and multi-step retrosynthesis. Notably, this was achieved with minimal training time, limited resources, and without modifications to the default AiZynthTrain settings.

Our approach, based solely on Machine Learning and Deep Learning principles, treats the retrosynthesis problem as a classification challenge. We believe that this method can also be adapted and applied to other template-based approaches to enhance their performances.

## 5 FUTURE WORKS

We recognize that our exploration of this field is just beginning. Notably, we have not yet transformed reaction templates to reduce the number of templates requiring classification—a crucial step demonstrated in other top-performing studies to enhance performance.

Future work will involve training a model on larger datasets such as Reaxys, and integrating USPTO-FULL with synthetic reaction datasets. Additionally, we plan to delve deeper into chemical and molecular structures to transform the dataset, including optimizing the structure of reaction templates to reduce the actual number of templates necessary for classification. We also aim to adapt our approach to current State-of-the-Art models like: LocalRetro, GLN to evaluate whether our approach can improve their performances.

### ACKNOWLEDGMENTS

We would like to acknowledge the contributions of the authors of AiZynthFinder, AiZynthTrain, RXNMapper, and Reaction Utils for making these valuable tools open-source, which greatly facilitated solving synthesis problems.

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

# A  APPENDIX

## A.1  SUBCLASS MAPPING

The number of clusters per class is an important hyperparameter that needs to be fine-tuned because a small number of clusters per class will lead to a small variance (which might cause underfitting), while a large number will provide a small bias (but might cause overfitting).

In our experiments, we set the number of clusters by $\left\lceil \frac{n}{k} \right\rceil$, where $n$ is the number of samples in a given class, and $k$ is a constant ranging from 50 to 500. We observed the results after the fine-tuning process: a small number of clusters (i.e., a large $k$) is usually better on the validation and test sets. On the other hand, a large number of clusters (i.e., a small $k$) can give better results on the training set but slightly worse results on the validation and test sets.

We also explored whether using subclass mapping before training the backbone model could be beneficial. This approach is intriguing because it could potentially mitigate the impact of dataset imbalance and improve the embedding process of the backbone model.

However, it is important to note that the input consists solely of molecular fingerprints, which are numerical representations of molecules. These fingerprints may not provide significant information for the subclass mapping process. Our test of this approach resulted in poorer performance compared to our initial method. This notably worse result highlights the importance of a well-defined projection of our Metric Transform model to the embedding space and underscores the need for an effective projection to enhance the utility of the subclass mapping.

To further explore the utility of subclass mapping, we conducted additional experiments:

1. Set the limit of samples for subclasses.

2. Fine-tune the backbone model on the subclassed dataset.

In experiment 1, we observed a slight improvement, making it easier to achieve the results shown in Table 1.

In experiment 2, the final results significantly improved, with performance gains ranging from 0.1 to 0.5 for top-1 to top-50 accuracy.

The reason behind these improvements is that, after subclass mapping, samples from the same class may belong to different subclasses while still sharing characteristics from the original class. Fine-tuning the backbone model on the new subclassed dataset allows for better separation of these subclasses.

Please note that the results on one-step and multi-step retrosynthesis presented above were obtained without incorporating this new exploration.

## A.2  RING-BREAKER MODEL

The idea behind the ring-breaker model is to help the model break ring systems in a given molecule, thereby making it easier to decompose that molecule into smaller and purchasable precursors (Thakkar et al. 2020).

The mixed results obtained by incorporating the ring-breaker model into multi-step retrosynthesis (see Table 6) can be explained by the following elements:

1. Search algorithms need to allocate separate time for evaluating both the one-step model and the ring-breaker model, which reduces effectiveness when the search time is limited.

2. The ring reaction dataset is not large enough to effectively cover ring systems.

These two points lead to an ineffective situation where the one-step model is already sufficient for solving small and medium molecules, but large molecules remain unsolvable (see Table 6, 8 and 9). Extensive research is needed to effectively incorporate ring-breaker models into multi-step retrosynthesis planning.

### A.3 ANALYSIS OF THE SUBSET OF 60 MOLECULES OF BAYER'S CHEMISTS AND PUBCHEM

We provide a comprehensive analysis of the time required to solve each molecule in the subset of 60 molecules sourced from Bayer's chemists and PubChem. We note that solving time is highly dependent on computational resources, particularly for more challenging molecules. Therefore, this analysis aims to highlight the differences in performance between MCTS and NMCS.

| ID | Time (s) | ID | Time (s) | ID | Time (s) | ID | Time (s) |
|------|------------|------|------------|------|------------|------|------------|
| A0 | 73.1/274.3 | A1 | 0.02/1.25 | A2 | 0.06/21.7 | A3 | 147.9/416.1 |
| A4 | 27.4/213.1 | A5 | 558.6/252.5 | A6 | X/1035 | A7 | 131.7/165.4 |
| A8 | X/1404 | A9 | 6.6/13.5 | A10 | X/1420 | A11 | 0.06/4.5 |
| A12 | 98.6/184 | A13 | X/1539 | A14 | 0.06/14.8 | A15 | 0.9/41.6 |
| A16 | 0.3/11.7 | A17 | 0.01/33.6 | A18 | 5.2/116.6 | A19 | 532.6/7.3 |
| C1 | 0.02/1.3 | C2 | X/X | C3 | 0.03/3.4 | C4 | X/X |
| C5 | 0.02/1.3 | C6 | X/X | C7 | X/X | C8 | 0.03/2.9 |
| C9 | X/X | C10 | X/X | C11 | X/X | C12 | X/X |
| C13 | 0.04/2.5 | C14 | 0.01/32.4 | C15 | X/X | C16 | X/X |
| C17 | X/X | C18 | X/X | C19 | 22/46.6 | C20 | 10.8/1514.6 |
| C21 | 0.09/2.7 | C22 | X/647.9 | C23 | 0.03/1.6 | C24 | X/X |
| C25 | 0.01/1.3 | C26 | 0.02/1.27 | C27 | X/X | C28 | X/X |
| C29 | X/X | C30 | X/X | C31 | 0.01/32.2 | C32 | X/X |
| C33 | X/X | C34 | 0.03/2.9 | C35 | X/2217 | C36 | 1.7/21.1 |
| C37 | 13.3/81.6 | C38 | X/X | C39 | X/X | C40 | 0.01/33.9 |

Table 9: Analysis of the time required (in seconds) to solve each molecule in the subset of 60 molecules from Bayer's chemists and PubChem using MetricTransform. The values to the left of the slash represent times from Monte Carlo Tree Search, while those to the right correspond to times from Nested Monte Carlo Search. 'X' indicates non-solvable molecules within the time limit of 3600 seconds.

Our analysis indicates that, using the same one-step retrosynthesis model, NMCS generally demonstrates superior performance compared to MCTS. Every molecule successfully solved by MCTS was also addressed by NMCS, although NMCS required more time. One potential strategy is to use MCTS to solve the "easier" molecules first, followed by NMCS for the more challenging ones. This approach would allow us to benefit from the speed of MCTS while leveraging the higher solving performance of NMCS.

We also highlight the challenges of multi-step retrosynthesis, particularly with large molecules such as C2 and C39. These molecules contain complex ring systems that remain unsolvable in our experiments. One potential strategy is to employ a ring-breaker model to simplify the retrosynthesis planning by breaking down these ring systems. However, as discussed in Section A.2, the USPTO-FULL ring breaker model has drawbacks that affect the solving capacity of our model and search algorithms for these large molecules.

Additionally, we encounter difficulties with some medium-sized molecules, such as C6 and C7, which also remain unsolvable. This issue is likely due to the lack of specific templates needed to break down these molecules effectively.

Interestingly, we found that C22 and C38, which remain unsolved using MCTS, have precursors that are not in stock—Br for C22 and O=C1OC(=O)C2C3C=CC(C3)C12 for C38—but these precursors can be easily purchased or found on eMolecules. This highlights the importance of having

a large dataset and a detailed stock database to enhance the ability to solve multi-step retrosynthesis problems.

