# OpenReview forum: "Application of Metric Transformation in One-Step Retrosynthesis"
_ICLR.cc/2025/Conference — Submitted to ICLR 2025_

### Official Review · Reviewer_jLnV · 2024-11-01

**Soundness:** 2
**Presentation:** 2
**Contribution:** 2
**Rating:** 5
**Confidence:** 5

**Summary:**

In this work, the author addresses template imbalance in inverse synthesis by introducing a hierarchical classification approach. Besides, they use k-means clustering to split large template classes into subclasses and Implement proxy anchor loss for classification. Moreover, subclasses were mapped back to the original template categories.

**Strengths:**

The research effectively addresses a practical challenge in the field - the imbalance of template distributions in datasets. The approach demonstrates measurable improvements on the USPTO full dataset, offering a practical solution to reduce intra-class variance. The methodology shows progression in handling imbalanced data, with a clear focus on improving classification accuracy through hierarchical structuring.

**Weaknesses:**

Despite its contributions, the paper suffers from several limitations. The methodology presentation lacks clarity and precision, requiring interpretation to understand the core concepts. While improvements are shown, the achieved 50% accuracy falls behind the current state-of-the-art performance. The paper also omits comparisons with recent superior methods, raising questions about its relative contribution to the field. Additionally, the overall writing style lacks precision and clarity, making it challenging to fully grasp the technical details without considerable effort.

**Questions:**

What is the computational overhead introduced by the k-means clustering step, and how was the optimal number of subclasses determined?
Could the approach be enhanced by implementing more sophisticated clustering methods beyond k-means?
How does this method perform when applied to datasets other than USPTO full?

---

> ### Author Response · Authors · 2024-11-13
>
> We thank the reviewer for their constructive feedback.
>
> **Regarding the weakness:**
> - Our primary goal is to achieve fast inference with strong predictive capacity to support multi-step retrosynthesis. To support this, we address the dataset's inherent imbalance to improve classification accuracy while maintaining the inference speed. Additionally, we observed that using the same architecture of our model, but without the imbalance-handling methods (proxy anchor loss and subclass mapping) yielded results nearly identical to those of the less performant AZT model, even when attention layers were stacked.
>
> **Regarding the question:**
>
> - For the k-means clustering step, it took us less than an hour for the entire USPTO-full dataset, which is quite fast, even though we also applied it to the small/medium classes, which were not required for our proposed solution.
> - The optimal number of subclasses depends on the logic of the remap layer that we applied. However, we found that setting the number of subclasses to 6 is a good choice because it reduces the imbalance between classes while still maintaining good performance when remapped back to the original class using the max between its subclasses.
> - In Appendix A.1, we describe some enhancements that improve performance, but we have not yet experimented with other clustering methods.
> - We have only applied our method to the USPTO-full dataset and its subsets. We observed that the more severe the imbalance, the better the performance of our proposed method. However, other datasets are typically private, making it difficult to gain further insight.
>
> **Further discussion:**
> - We believe that the results of the multi-step retrosynthesis tasks are good, both in terms of speed and the number of solved molecules, even with a significantly smaller dataset, which aligns with our main goal.
> - The imbalance-handling methods we used (proxy anchor loss and subclass mapping) are independent of the specific architecture, meaning they can be applied to other state-of-the-art template-based models to enhance their performance on imbalanced datasets, which is something we plan to explore in future work.

---

> > ### Comment · Reviewer_jLnV · 2024-11-25
> >
> > I appreciate the author's reply. I will maintain my rate.

---

### Official Review · Reviewer_5UY6 · 2024-11-03

**Soundness:** 1
**Presentation:** 1
**Contribution:** 1
**Rating:** 1
**Confidence:** 5

**Summary:**

Retrosynthesis is a central challenge in chemistry, crucial for effectively leveraging newly discovered and complex molecules. In this article, the authors present a framework that combines proxy anchor loss with a depth metric Transformer to improve retrosynthesis accuracy in template-based methods, framing the inverse synthesis problem as a classification task. Although the results demonstrate some improvement, the method lacks novelty and does not achieve optimal accuracy.

**Strengths:**

None

**Weaknesses:**

1. Retrosynthesis accuracy is low: Compared with state-of-the-art models such as LocalRetro, the retrosynthesis accuracy is low, performing poorly across multiple datasets.
2. Lacks comparative experiments: There is a lack of comparative experiments to demonstrate the improvement achieved by the applied method, making it difficult to see the advantages of this approach.
3. Limited diversity and practicality: The model is restricted to template-based methods and cannot be extended to a broader chemical space.
4. Inconsistent writing format: The manuscript's format presents issues,  such as the abstract, which lacks clear segmentation into distinct sections. This impacts the clarity and structured presentation of key information, making it challenging for readers to discern the main contributions and findings.

**Questions:**

1. The comparison lacks certain key methods, such as RetroExplainer and RetroKNN, and the reported results do not achieve state-of-the-art performance on the USPTO-50k dataset.

2. In Appendix A.2, only the limitations of the ring break model are discussed; however, its functional role in the model is not clearly explained.

3. Please clarify the distinction between the proxy anchor method employed and a clustering approach. Additionally, why is the same approach not applicable for segmenting larger subsets?

4. Section 3.1 assumes that the additional 11,567 data points are all failures for both our model and the baseline. Why are some of these not classified as correct predictions, at least in proportion?

5. Why was the inference speed of different models not benchmarked on the same device to ensure fair comparison?

6. While template coverage has improved, the results do not surpass those in previous studies. Could you provide an explanation for this outcome?

7. Are there corresponding ablation studies to validate the contribution of each component in the proposed framework? How do you explain the significant discrepancies between the results in Table 5 and those of the baseline model?

8. Minor formatting issues include a missing period at line 182, an extraneous period at line 278, and an extra space before the period on line 433.

---

> ### Author Response · Authors · 2024-11-13
>
> We thank the reviewer for their feedback.
>
> **Regarding the weakness:**
>
> - Our primary goal is to achieve fast inference with strong predictive capacity to support multi-step retrosynthesis. To achieve this, we address the dataset's inherent imbalance to improve classification accuracy while maintaining inference speed, which is our main contribution. Additionally, we reconfirm the effectiveness of Nested Monte Carlo search for multi-step retrosynthesis.
> - In the multi-step retrosynthesis section, we use AiZynthFinder, a template-based synthesis tool. This limited our work to this scenario. Moreover, we did not propose a novel model but rather a solution to tackle the class imbalance, which is a challenge present in many domains.
>
>
> **Regarding the question:**
>
> 1. We tackle the imbalance in the dataset to achieve better performance using the same model architecture (in our paper, this is a vanilla attention layers). Due to the independence of the imbalance-handling methods (proxy anchor loss and subclass mapping) from the model architecture, we can apply these methods to other state-of-the-art architectures, which we plan to explore in future work.
> 2. The ring-breaking model of AiZynthTrain is inspired by the work of Thakkar et al. (2020), where the model breaks the ring system to simplify the given molecule.
> 3. Proxy Anchor loss is used to train the backbone model to distinguish the similarities and differences between samples from the same class or different classes before fine-tuning the model to predict the correct class (template). On the other hand, subclass mapping (a clustering approach) is used to reduce the imbalance between classes, improving performance on small and medium-sized classes. Together, these methods effectively learn the representation of the given molecule and reduce imbalance. As a result, this combination works better on larger datasets compared to smaller ones, as shown in Table 1 (USPTO-full) and Table 5 (USPTO-50k).
> 4. We maintain the same ratio as the default to ensure a fair comparison on the multi-step retrosynthesis task. We thus, consider the worst-case scenario, where all these data points are considered failures to predict.
> 5. Some models reported in Seidl et al. (2022) use significantly stronger GPUs than ours. Moreover, we wanted to test whether the proposed imbalance-handling methods would slow down inference speed. As shown in the paper, this effect is not significant.
> 6. The coverage result is based on the fact that some reaction rules are absent from the training set. In other papers, such as LocalRetro and GLN, coverage (excluding reaction rules in the testing set but not in the training set) is often used as a theoretical upper bound. We propose this measure to show that, even though our model does not surpass state-of-the-art models, it still achieves good performance, even as the number of reaction rules grows.
> 7. We also conducted ablation studies:
>    - If we remove the imbalance-handling methods and train our model as usual, the result shows only a slight improvement (about 0.1-0.5%) compared to the baseline, even when stacking the attention layers.
>    - The subclass mapping (clustering approach) works well on large datasets, where there is significant class imbalance.
>    - In Table 5, we show the improvement of our approaches even on small datasets, compare to baseline. However, it falls behind state-of-the-art models, which we find reasonable, as our proposal is purely focused on tackling the imbalance among classes.
>
> **Further discussion:**
>
> - We believe that the results of the multi-step retrosynthesis tasks are good, both in terms of speed and the number of solved molecules, even with a significantly smaller dataset, which aligns with our main goal.

---

> > ### Comment · Reviewer_5UY6 · 2024-11-28
> >
> > Thank you for your responses. I appreciate your efforts, but I still have some concerns:
> >
> > Question 1: Accuracy
> > Your method has lower accuracy than models like LocalRetro. I understand your focus on class imbalance, but it would help to see a direct comparison or explanation of the performance gap.
> >
> > Question 2: Comparisons
> > There are no comparisons with key methods like RetroExplainer or RetroKNN. Without these, it is hard to understand the advantage of your approach.
> >
> > Thank you again for your explanations. I encourage you to address these points further. I will maintain my score.

---

### Official Review · Reviewer_oPn6 · 2024-11-03

**Soundness:** 1
**Presentation:** 1
**Contribution:** 1
**Rating:** 1
**Confidence:** 4

**Summary:**

The article explores using Deep Metric Learning and Transformer architecture for predicting chemical compound retrosynthesis in SMILES format. It shows that combining Attention mechanisms with Proxy Anchor Loss improves classification by capturing local and global contexts, achieving promising accuracy on the USPTO-FULL dataset without requiring prior chemical knowledge.

**Strengths:**

The use of Attention mechanisms and Proxy Anchor Loss enhances the performance of classification tasks by capturing both local and global contexts.

**Weaknesses:**

The retrosynthesis accuracy is relatively low compared to state-of-the-art models.
The model is restricted to template-based approaches, which hinders its applicability and scalability across a broader spectrum of chemical environments.

**Questions:**

Why are key methods like RetroExplainer excluded from the comparison？
How do your results compare to the state-of-the-art on the USPTO-50k dataset?

---

> ### Author Response · Authors · 2024-11-13
>
> We thank the reviewer for their feedback.
>
> **Regarding the question:**
>
> - Our results fall behind state-of-the-art models such as LocalRetro, OGNN, and RetroExplainer on USPTO-50k. However, we show significant improvement compared to the baseline and the same architecture model, trained without the imbalance-handling methods (proxy anchor loss and subclass mapping) that we proposed.
>
> **Further discussion:**
>
> - We propose a solution to address the dataset's imbalance, aiming to improve classification accuracy while maintaining inference speed—this being our main contribution.
> - The imbalance-handling methods we used are independent of the specific architecture, meaning they can be applied to other state-of-the-art template-based models to enhance their performance on imbalanced datasets, which we plan to explore in future work.
> - We believe the results of the multi-step retrosynthesis tasks are promising, both in terms of speed and the number of solved molecules, even with a significantly smaller dataset. This aligns with our main goal.

---

> ### Author Response · Authors · 2024-11-13
> **Additional response**
>
> Hello reviewer,
>
> You can check Table 5 in the paper, where I compare the models on the USPTO-50k dataset. However, as stated in the paper, we focus more on the USPTO-Full dataset because our primary goal is multistep retrosynthesis.

---

### Meta-Review · Area_Chair_Yj78 · 2024-12-15

**Metareview:**

The paper explores Deep Metric Learning and Transformers for chemical retrosynthesis in SMILES, using Attention and Proxy Anchor Loss to improve classification. It shows some accuracy improvements on the USPTO-FULL dataset but lags behind state-of-the-art models.

Strengths: Addresses template imbalance in retrosynthesis; Combines Attention mechanisms with Proxy Anchor Loss for enhanced classification.

Weaknesses: Lower accuracy compared to state-of-the-art;  Lacks clarity in methodology presentation; Omits comparison with recent superior methods.

The paper is rejected due to its lower performance compared to state-of-the-art, lack of comparative analysis with key methods, and unclear presentation. Further work is needed to improve accuracy and clarity.

**Additional Comments On Reviewer Discussion:**

All the reviewers raised that the performance of the proposed algorithm is lower than SOTA methods, and the authors should compare their algorithm with recent superior methods. The authors does not address these concerns.

---

### Decision · Program_Chairs · 2025-01-22

Reject